# The prevalence of and factors associated with depressive and anxiety symptoms during the COVID-19 pandemic among healthcare workers in South Africa

**Megan Pool**[1]☯*, **Katherine Sorsdahl**[1]☯, **Bronwyn Myers**[2,3,4]☯, **Claire van der Westhuizen**[1]☯

**1** Alan J. Flisher Centre for Public Mental Health, Department of Psychiatry and Mental Health, University of Cape Town, Cape Town, South Africa, **2** Curtin enAble Institute, Curtin University, Perth, Australia, **3** Department of Psychiatry and Mental Health, University of Cape Town, Cape Town, South Africa, **4** Mental Health, Alcohol, Substance Use and Tobacco Research Unit, South African Medical Research Council, Cape Town, South Africa

☯ These authors contributed equally to this work.
* megan.pool@uct.ac.za

**Data Availability Statement:** The data that support these findings are owned by the Western Cape Department of Health and applicants may apply

## Abstract

### Introduction

Healthcare workers globally have experienced increased social and occupational stressors in their working environments and communities because of COVID-19 which has increased the risk of mental health concerns. This study aimed to explore the prevalence and correlates of depression and anxiety amongst healthcare workers during the COVID-19 pandemic in the Western Cape, South Africa. In addition, role-related stressors and coping strategies were explored.

### Material and methods

We conducted a cross-sectional survey of doctors and nurses working in public healthcare facilities across the Western Cape, South Africa. Participants completed the Generalized Anxiety Disorder-7 (GAD-7), the Center for Epidemiologic Studies Depression (CES-D), the Professional Quality of Life (PROQL-R-IV), and the Brief Coping Orientation to Problems Experienced (COPE-R) scales. Data were analysed using multivariable logistic regression analysis.

### Results

The sample comprised 416 health workers (303 nurses, 113 doctors). Almost 40% of the sample (n = 161) had CES-D scores suggestive of probable depression, and 45.9% (n = 186) had GAD-7 scores suggestive of anxiety. In the logistic regression model, the odds of probable depression were higher for female participants compared to men (OR = 2.26, 95% CI 1.00–5.10) and for participants who used behavioural disengagement as a coping strategy (OR = 1.50, 95% CI 1.14–1.97). More time spent working with COVID patients was

online to the National Health Research Database (https://nhrd.health.gov.za/).If you have any technical related queries, please send a mail to NHRD@Health.gov.za.

**Funding:** The Degree from which this study emanated was funded by the South African Medical Research Council (SAMRC) through its Division of Research Capacity Development under the Bongani Mayosi National Health Scholars Programme from funding received from the Public Health Enhancement Fund/South African National Department of Health. The content hereof is the sole responsibility of the authors and does not necessarily represent the official views of the SAMRC or the funders. Funding bodies has no role in the research activity. All authors were independent from the funders and had access to the study data. The funders had no role in study design, data collection and analysis, decision to publish, or preparation of the manuscript.

**Competing interests:** The authors have declared that no competing interests exist.

associated with increased odds of having high levels of anxiety [OR = 1.13, 95% CI (1.02–1.25). Substance use (OR = 1.39, 95% CI 1.08–1.81), venting (OR = 1.31, 95% CI 1.01–1.70), and self-blame (OR = 1.42, 95% CI 1.08–1.87) were some of the coping strategies used by healthcare workers. High levels of secondary traumatic stress and burnout were found to increase the odds of both depression and anxiety.

## Conclusion

Findings of this study suggest that there is a high prevalence of mental health issues among healthcare workers, and a critical need to focus on workplace mental health interventions to support these frontline workers.

## Introduction

Although communities globally experienced widespread hardship and mental health difficulties during the COVID-19 pandemic [1–3], this was particularly the case for frontline healthcare workers who endured extraordinary pandemic-related challenges [4–7]. Several studies have highlighted the mental health concerns of healthcare workers during the pandemic [8–10]. For example, a meta-review examining the impacts of COVID-19 on healthcare workers' mental health found that anxiety, depression, and post-traumatic stress disorder (PTSD) were the most prevalent conditions [9]. However, healthcare workers are known to be a vulnerable group regarding mental health symptoms and challenges in non-pandemic circumstances.

Most pre-pandemic studies on mental health concerns among healthcare workers were conducted in high-income country settings and focused on particular disciplines, such as emergency medicine or oncology [11, 12]. There is also some evidence from low- and middle-income countries (LMIC) that mental health concerns are prevalent among generalist healthcare workers [13] This is not surprising since LMICs, such as South Africa, have fragile and under-resourced public health systems [14] with healthcare workers working under high pressure and in poor conditions [15, 16]. The mental health needs of healthcare workers were highlighted in one example where the investigators reported high levels of depression and associated burnout among 132 doctors working in South African public healthcare settings with 30% reporting symptoms of moderate or severe depression prior to the pandemic [17].

The COVID-19 pandemic appears to have significantly impacted and foregrounded concerns about the mental health of this already strained workforce. A systematic review of studies conducted among South African healthcare workers during the pandemic reported that between 7.4% and 35.0% of frontline healthcare workers experienced symptoms of post-traumatic stress disorder (PTSD), anxiety, and/or depression [18]. Similarly, Dawood, Tomita (2022) [19] found high prevalence of depression, 51.5%, anxiety, 47.2% and stress, 44.43% among healthcare workers working in KwaZulu Natal Province, South Africa.

When left undetected and unaddressed, mental health concerns among healthcare workers hold a myriad of implications. For example, mental health problems among healthcare providers impact on the efficiency of the public health system (due to lost productivity) and the quality of patient care [20–22]. Two of the main drivers of lost productivity are absenteeism and presenteeism. Absenteeism, or the absence from work due to a physical or psychological condition, is known to lead to staff shortages and impact productivity [23]. Presenteeism, or the practice of being physically present at work when unwell, impacts on productivity and quality of work [20, 24]. Individuals with mental health concerns may be particularly prone to

presenteeism [25]. While the focus of healthcare workers has primarily been on attending to the distress and trauma experienced by others, the act of service and helping may have both positive and negative implications for their own mental health [26, 27]. Mental health concerns among healthcare workers may also impact on care quality, affecting waiting times for care, quality of provider-patient interactions, and leading to more adverse events and missed diagnoses [28, 29]. Thus, it is vital that risk and protective factors for mental health concerns be identified in these workers.

During the pandemic, healthcare workers were exposed to increased social and occupational risk factors for mental health problems, such as fatigue, stigma, secondary trauma, fear of being exposed to the virus, and fears of exposing others [4, 30, 31]. Compounding these factors, some research suggests that healthcare workers are hesitant and unwilling to seek help [32, 33]. Data is scarce regarding other potential risk factors, such as coping strategies used by frontline workers. A few international studies have explored healthcare workers' use of coping strategies but not associations with mental health symptoms [34]. One study conducted in Malaysia with 137 healthcare workers found that active coping and acceptance were associated with decreased depressive symptoms while no strategies seem to be protective against anxiety symptoms [35].

Understanding role-related and personal factors associated with the risk of depression and anxiety amongst healthcare workers is a vital first step toward identifying strategies to support healthcare workers in times of crisis. This study aims to address this gap by exploring the prevalence and factors associated with depression and anxiety amongst healthcare workers during the COVID-19 pandemic in the Western Cape, South Africa. Although a lot of research has been conducted on the impacts of COVID-19 on healthcare workers, this study is among the first to look at healthcare workers' strategies for coping and their mental health help-seeking intentions.

## Methods

This quantitative cross-sectional research study was informed by previous qualitative work in primary healthcare clinics which documented the multiple stressors experienced by healthcare workers and high levels of burnout in the workplace [36].

### Study settings

We recruited healthcare workers from a range of primary healthcare facilities as well as district, regional, and tertiary hospitals across the Western Cape. We approached thirty-five healthcare facilities (clinics and hospitals) and thirty responded with permission to take part in the online survey. Of the thirty facilities that granted permission for in-person data collection or delivery of hard copy questionnaires, only twenty-three took part in the study. The survey was distributed electronically; therefore, additional facilities were represented in the data. Furthermore, ethical approval for this study was obtained from the University of Cape Town's Human Research Ethics Committee 166/2020. In addition, ethical approval was obtained from the Western Cape Department of Health (WC_202007_053) as well as the respective operational managers, and heads of departments for each hospital or clinic. Written informed consent was obtained for each participant. Three tertiary hospitals were included, six district hospitals, three regional hospitals and eleven primary healthcare facilities.

### Participants

Healthcare workers were eligible to participate in the study if they were working in the Western Cape, registered with either the Health Professions Council of South Africa (HPCSA) or

the South African Nursing Council (SANC), and worked in either a clinical role and/or a managerial position as a nurse or a doctor. A total of 452 healthcare workers completed the self-administered questionnaire, of which 36 responses were removed because participants were not eligible based on their profession or location, leaving a total of 416 responses. Recruitment of participants occurred over an 18-month period (July 2020- February 2022). The first author contacted the respective facilities' managerial staff to explain the purpose of the study. Electronic information sheets about the study were provided. Managerial staff distributed a link to an electronic survey (including information sheets and consent forms) to potential participants via email. At various points in the pandemic, when permitted, the first author distributed hard copies of the consent forms and questionnaires at participating facilities and arranged to collect completed forms. Where permitted by the country's COVID-19 restrictions, the researcher attended small clinical meetings at participating facilities to introduce and recruit for the study. All participants provided informed consent prior to completing the survey.

## Ethical considerations

All procedures contributing to this study conform with the ethical standards of the Helsinki Declaration of 1975, revised in 2008. The following ethical standards to safeguard the well-being and rights of the participants were considered. Written informed consent was obtained for each participant. The principles of privacy and confidentiality were maintained, ensuring that sensitive information remained secure and protected. Participants were informed about the voluntary nature of their involvement in the study, and their consent was obtained prior to any data collection or analysis. Furthermore, ethical considerations extended to providing participants with avenues for seeking assistance or expressing distress, such as offering contact information for support services.

## Measures

**Sociodemographic and employment factors.** Information was collected regarding the participant's age; level of education; marital status; current employment status and role; length of time in the current role; place of work including the type of facility and department. We also included questions on sources of support available during the pandemic.

**Depression.** The Centre for Epidemiological Studies Depression Scale (CESD-10) was included to measure depressive symptoms [37]. A cut-off score of 10 and above is commonly used to identify individuals with significant depressive symptoms, as an indicator of possible depression. This measure has been used in various settings and validated in the South African context [38]. In the South African validation study the CESD-10 was statistically significantly positively correlated with the patient health questionnaire (PHQ-9), showing acceptable concurrent validity. The area under the receiver operating curve (ROC) curve was above 0.80 showing excellent criterion validity using the mini-international neuropsychiatric interview V6.0 depression module as the gold standard. The Cronbach's alpha ranged between 0.69 and 0.89 indicating acceptable internal consistency.

**Anxiety.** The Generalised Anxiety Disorder Scale-7 (GAD-7) is a self-rated screening tool and indicator for moderate and severe symptoms of anxiety and was included in the survey. This measure has good reliability and validity [39]. A cut-off score of 10 is commonly used to identify individuals with moderate-high anxiety symptoms [40]. The area under the ROC was 0.86 showing excellent criterion validity using the using the generalised anxiety disorder module of structured clinical interview of the Diagnostic Statistical Manual version IV (DSM-IV). The GAD-7 displayed excellent internal consistency with a Cronbach's alpha of 0.87.

**Role-related characteristics.** The Professional Quality of Life Scale (PROQL-R-IV) [41] is the most frequently used measure of the adverse and positive effects of helping others who experience distress and trauma. The ProQOL has three sub-scales for compassion satisfaction, burnout, and secondary trauma. Burnout; is a state of physical, emotional, and mental exhaustion that results from chronic workplace stress [42]. Secondary trauma refers to a state of distress or trauma that's indirectly experienced by hearing details of or witnessing a traumatic experience by another person [43]. Compassion satisfaction is a satisfying feeling that one experiences when helping others, it's the pleasure derived from doing ones work [44]. These subscales are categorised as low (scores 0–22), moderate (scores 23–41), and high (scores ≥42). For burnout and secondary trauma scales, we re-coded the scores into two categories, low (0–22) and moderate/high (23–50) as there were too few participants in the high category only. For the compassion satisfaction scale, given that only four participants scored in the low category, we combined participants in the low and moderate categories into a single low-moderate category. According to Stamm (2010), the ProQOL has demonstrated good construct validity [41]. In addition, the ProQOL has reported a Cronbach's alpha reliability ranging from 0.84 to 0.90 on three subscales [41].

**Coping and help-seeking.** The Brief Coping Orientation to Problems Experienced (COPE-R) measure was used to assess how healthcare workers manage stressors in their life. This self-administered multi-dimensional inventory was developed by [45] to identify coping strategies in response to distress. The Brief-COPE has 28 statements that measure helpful and unhelpful ways to cope with stressful life situations. The measure has illustrated good reliability and concurrent validity analyses indicated that these factors align well with self-efficacy for different types of coping [46, 47]. It measures the following three domains: 1) problem focused coping, 2) emotion-focused coping and 3) avoidant coping strategies through 14 subscales. These subscales are: Self-distraction (avoidant), Active coping (problem-focused), Denial (avoidant), Substance use (avoidant), Use of emotional support (emotion-focused), Use of instrumental support (problem-focused), Behavioural disengagement (avoidant), Venting (emotion-focused), Positive reframing (problem-focused), Planning (problem-focused) Humour (emotion-focused), Acceptance (emotion-focused), Religion (emotion-focused), and Self-blame (emotion-focused) Problem focused coping styles indicates strategies aimed at altering a stressful situation. Emotion focused coping styles aim to regulate emotions associated with the stressor and avoidant coping strategies indicate efforts to disengage from the stressor [45, 48]. The General Help-Seeking Questionnaire (GHSQ) was administered to formally assess help-seeking intentions and behaviours amongst healthcare workers. The scale has demonstrated satisfactory reliability and validity [49]. In a previous study examining the psychometric properties of the GHSQ they determined a Cronbach's alpha of 0.85 and a test-retest reliability assessed over a three-week period of 0.92 [49].

**COVID-19 impact questionnaire.** The Pandemic Stress Index is a three-item measure of stress and behaviour changes that individuals may experience during the COVID-19 pandemic [5]. To our knowledge the psychometric properties of this scale have not yet been investigated.

## Sample size calculation

We based our sample size calculation on the primary hypothesis: those people experiencing burnout or secondary traumatic stress will be more likely to have depression. We set the power at 80% and the significance level at 0.05. Based on the literature we used a conservative estimate that 30% of those experiencing STS or burnout would be experiencing high levels of depressive symptoms. We used open Epi to calculate the sample size for this cross-sectional study [50]. For adequate power, it was determined that a total of 291 healthcare workers would be sufficient for this cross-sectional study.

## Data analysis

The Statistical Package for the Social Sciences (SPSS, version 27) was used for analysis. Frequency distributions and descriptive statistics were calculated for categorical and continuous variables. Descriptive statistics were conducted and compared across groups by profession (doctors/nurses) using chi-square tests. Separate logistic regression models were developed for depression and anxiety. The models explored the unadjusted and adjusted associations between sociodemographic, work-related characteristics, coping, help seeking intentions and pandemic related stress, and depression and anxiety, respectively. For the adjusted models, variables that were significant in the unadjusted model in addition to age and gender were included. The findings are reported as odds ratios (OR) with 95% confidence intervals (CIs). Statistical significance was based on 2-sided tests and set at $\alpha = 0.05$.

## Results

### Socio-demographic characteristics of the sample

Table 1 depicts the demographic profile of the healthcare workers. In total, 303 nursing professionals and 113 doctors were enrolled. More than half of the participants were in a relationship (n = 245, 59.2%). Most of the sample were female (n = 330, 79.7%) and between 20 and 35 years old 177 (42.9%). Just over half the participants (n = 204, 50.5%) were from tertiary hospitals, with the next largest group (n = 161, 40.1%) from primary healthcare clinics. Many participants (n = 314, 75.8%) were working in clinical positions. A high proportion of participants (n = 273, 68.9%) feared spreading the COVID-19 virus, with about two-thirds of these being nursing staff (n = 191, 67.3%).

### Mental health and psychosocial factors

A total of 161 (39.6%) healthcare workers obtained CES-D scores indicative of clinically significant depressive symptoms and 186 (45.9%) obtained GAD-7 scores suggestive of anxiety, with no significant differences found between the professions (Table 2). More than half of the participants scored in the moderate/high range on the burnout scale. A significantly higher proportion of doctors scored in the moderate/high range for burnout (65.2%) compared to nursing staff (50.5%), p = 0.01. Very high proportions of both professions reported low compassion satisfaction, with a significantly higher proportion of doctors (75%) reporting low satisfaction compared to nurses (56.1%), p = 0.001. Overall, a large proportion of the sample obtained scores indicative of high levels of secondary traumatic stress (n = 192; 47.3%).

Healthcare workers endorsed various acceptable sources of support for personal or emotional problems with family, intimate partners and friends being the most common choices (see Fig 1). The proportion of doctors and nurses who endorsed intention to seek support from intimate partners differed significantly (69.5% of nurses vs 90.2% of doctors, p = 0.001). A higher proportion of nurses reported the intention to seek help from religious advisors (n = 142, 23%) than doctors (n = 26, 50.9%; p = 0.001). Less than 50% endorsed seeking professional mental health support with (n = 241, 61.6%) indicating not having sought help from a mental health professional.

Various coping strategies were used by healthcare workers (see Fig 2). The most frequently used coping strategies were acceptance (n = 300, 75.9%), seeking religious leaders' support (n = 291, 73.9%) followed by active coping (n = 256, 64.2%), self-distraction (n = 254, 63.3%) and planning (n = 246, 61.7%). When the two groups of professionals were compared, a higher proportion of nurses than doctors used behavioural disengagement as a coping style (23.9% vs 15.2%: p = 0.06).

**Table 1. Sociodemographic and work-related characteristics of the sample by profession.**

| Demographics related characteristics | Total | Nurses | Doctors | P Value |
|---|---|---|---|---|
| | N = 416 (100%) | N = 303 (100%) | N = 113 (100%) | |
| **Gender** | | | | |
| Male | 84 (20.3%) | 32 (10.6%) | 52 (46.0%) | <0.001 |
| Female | 330 (79.7%) | 269 (89.4%) | 61 (54.0%) | |
| **Age** | | | | |
| 20–35 | 177 (42.9%) | 111 (37.0%) | 66 (58.4%) | <0.001 |
| 36–50 | 154 (37.3%) | 112 (37.3%) | 42 (37.2%) | |
| 51+ | 82 (19.9%) | 77 (25.7%) | 5 (4.4%) | |
| **Relationship status** | | | | |
| In a relationship | 245 (59.2%) | 162 (53.8%) | 83 (73.5%) | <0.001 |
| Single | 169 (40.8%) | 139 (46.2%) | 30 (26.5%) | |
| **Level of education** | | | | |
| Undergraduate | 124 (30.8) | 124 (42.8%) | 0 | <0.001 |
| Bachelor's Degree | 226 (56.1%) | 158 (54.5%) | 68 (60.2%) | |
| Postgraduate Degree | 53 (13.2%) | 8 (2.8%) | 45 (39.8%) | |
| **Place of work** | | | | |
| Primary Healthcare Clinics | 162 (40.1%) | 125 (42.5%) | 37 (33.6%) | 0.17 |
| District Hospitals | 38 (9.4%) | 29 (9.9%) | 9 (8.2%) | |
| Tertiary Hospitals | 204 (50.5%) | 140 (47.6%) | 64 (58.2%) | |
| **Length of time working** | | | | |
| <1 years | 35 (8.6%) | 26 (8.8%) | 9 (8.0%) | <0.001 |
| 1-5years | 191 (46.7%) | 115 (38.7%) | 76 (67.9%) | |
| 6–10 years | 87 (21.3%) | 74 (24.9%) | 13 (11.6%) | |
| 11–20 years | 63 (15.4%) | 50 (16.8%) | 13 (11.6%) | |
| 21–40 years | 34 (8.1%) | 32 (10.8) | 1 (0.9%) | |
| **Healthcare sector** | | | | |
| Public | 391 (94.0%) | 295 (97.4%) | 96 (85.0%) | <0.001 |
| Private | 12 (2.9%) | 7 (2.3%) | 5 (4.4%) | |
| Both | 13 (3.1%) | 1 (0.3%) | 12 (10.6%) | |
| **Role at work** | | | | |
| Clinical | 314 (75.8%) | 221 (73.4%) | 93 (82.3%) | 0.11 |
| Managerial | 24 (5.8%) | 21 (7.0%) | 3 (2.7%) | |
| Both | 76 (18.4%) | 59 (19.6%) | 17 (15.0%) | |
| **Work time spent working with COVID-19 patients (med, range)** | **3 (10)** | **4 (10)** | **3 (10)** | <0.001 |
| **Fear of getting COVID-19** | | | | |
| No | 175 (44.2%) | 115 (40.5%) | 60 (53.6%) | 0.02 |
| Yes | 221 (55.8%) | 169 (59.5%) | 52 (46.4%) | |
| **Fear of spreading COVID-19** | | | | |
| No | 123 (31.1%) | 93 (32.7%) | 30 (26.8%) | 0.25 |
| Yes | 273 (68.9%) | 191 (67.3%) | 82 (73.2%) | |

## Factors associated with depressive symptoms

Unadjusted and adjusted associations between sociodemographic, work-related, psychosocial characteristics and depression are shown in Table 3. In the adjusted model, four variables were significantly associated with depression. First, female participants had greater odds of depression than men (OR = 2.26, 95% CI 1.00–5.10). Second, the coping strategy, behavioural disengagement (OR = 1.50, 95% CI 1.14–1.97) was associated with increased odds of depression.

**Table 2. Mental health concerns, professional quality of life among participants (n = 416).**

| Mental Health Related Characteristics | Total | Nurses | Doctors | P value |
|---|---|---|---|---|
| | N = 416 (100%) | N = 303 (100%) | N = 113 (100%) | |
| **Depression (CESD-10)** | | | | |
| No | 246 (59.1%) | 173 (58.6%) | 73 (65.2%) | 0.23 |
| Yes | 161 (39.6%) | 122 (41.4%) | 39 (34.8%) | |
| **Anxiety (GAD-7)** | | | | |
| No | 219 (54.1%) | 157 (53.6%) | 62 (55.4%) | 0.75 |
| Yes | 186 (45.9%) | 136 (46.4%) | 50 (44.6%) | |
| **Compassion satisfaction (ProQOL)** | | | | |
| Low/moderate | 249 (61.3%) | 165 (56.1%) | 84 (75.0%) | 0.001 |
| High | 157 (38.7%) | 129 (43.9%) | 28 (25.0%) | |
| **Secondary traumatic stress (ProQOL)** | | | | |
| Low | 214 (52.7%) | 157 (53.4%) | 57 (50.9%) | 0.65 |
| Moderate/high | 192 (47.3%) | 137 (46.6%) | 55 (49.1%) | |
| **Burnout (ProQOL)** | | | | |
| Low | 183 (45.4%) | 144 (49.5%) | 39 (34.85) | 0.01* |
| Moderate/high | 220 (54.6%) | 147 (50.5%) | 73 (65.2%) | |

Finally, participants with high levels of secondary traumatic stress (OR = 6.57, 95% CI 3.30–13.04) and burnout (OR = 4.09, 95% CI 1.91–8.75) had increased odds of depression.

## Factors associated with anxiety symptoms

The unadjusted and adjusted associations between sociodemographic, work-related, and psychosocial characteristics, and anxiety are shown in Table 4. In the adjusted model, more time spent working with COVID-19 patients was associated with an increased odds of anxiety [OR = 1.13, 95% CI (1.02–1.25). Second, a few coping strategies were associated with greater odds of anxiety, specifically substance use (OR = 1.39, 95% CI 1.08–1.81), venting (OR = 1.31, 95% CI 1.01–1.70) and self-blame (OR = 1.42, 95% CI 1.08–1.87). Finally, participants experiencing high levels of burnout (OR =, 3.54 95% CI 1.82–6.99) and secondary traumatic stress (OR = 4.17, 95% CI 2.25–7.71) had greater odds of anxiety.

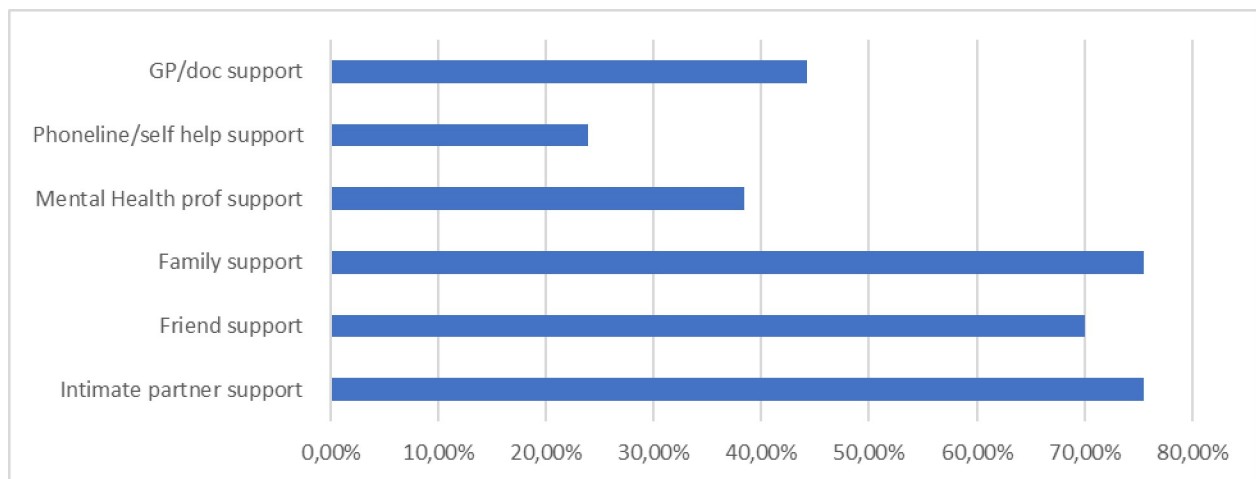

**Fig 1. General help seeking behaviours of healthcare workers.**

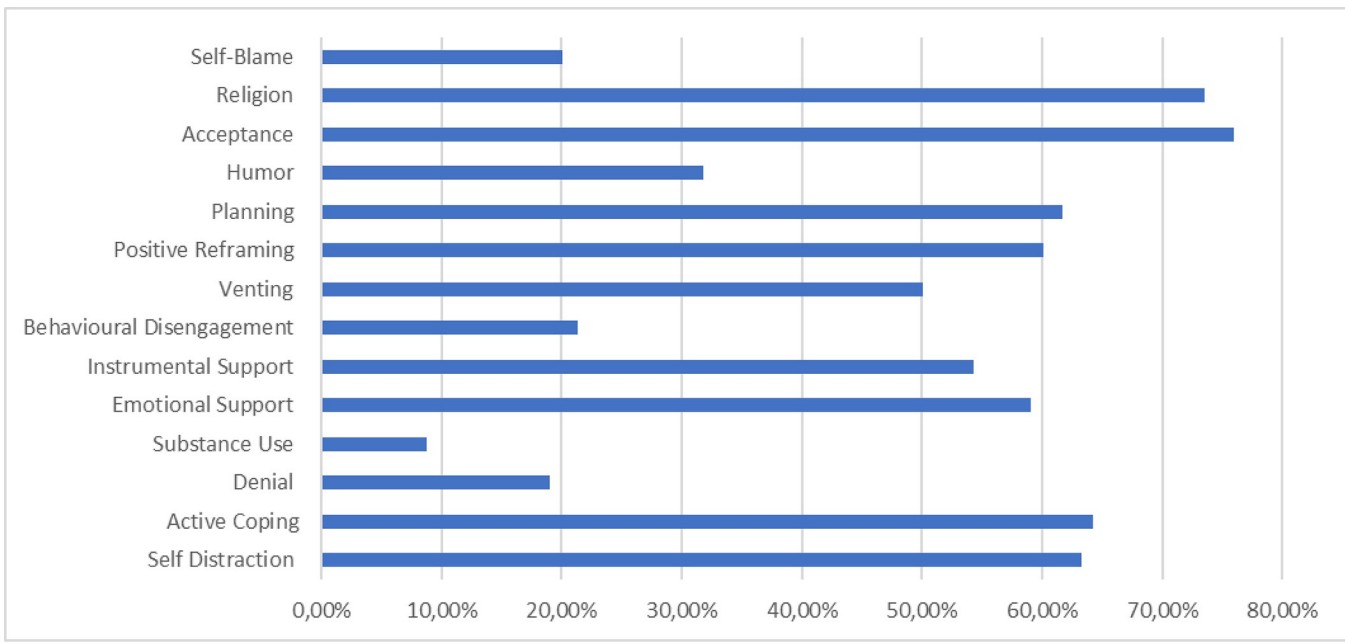

**Fig 2. Healthcare workers commonly used coping strategies.**

## Discussion

Several important findings emerged from this study. First, high levels of depression and anxiety symptoms were prevalent among healthcare workers working at healthcare facilities in the Western Cape during the COVID-19 pandemic. Second, healthcare providers who reported spending higher proportion of time spent working with COVID-19 patients were more likely to report probable anxiety than those with limited exposure. Third, several avoidant or maladaptive coping strategies were associated with probable depression or anxiety. Finally, high levels of secondary traumatic stress and burnout were found to increase the odds of both depression and anxiety.

First, in this study a high proportion of healthcare workers self-reported depression (N = 161, 39.6%) or anxiety symptom levels (N = 186 (46%) suggestive of risk for a mental health condition. Since the start of the COVID-19 pandemic in January 2020, special attention has been paid to the impact of the pandemic on mental health amongst healthcare workers globally and several studies have explored mental health symptoms in this group. A recent systematic review incorporating data from both LMICs, and high-income countries reported 24% prevalence of depression in healthcare workers as assessed by various screening tools [51]. Another systematic review found the pooled prevalence for depression and anxiety was 20.5% (95% CI 16.0%-25.3%) and 25.8% (95% CI 20.4%-31.5%) respectively. Our study reported a much higher prevalence. This study collected data from multi-sites using both online and paper-based surveys potentially accessing a high-risk group not captured by internet-based surveys. Our findings are in keeping with those from other local studies that also found high levels of depression amongst healthcare workers during the pandemic [19]. Since depression is linked to absenteeism, presenteeism and compromised patient care, this has serious implications, given that there are increased numbers of extremely unwell patients during pandemic periods.

Second, one of the potential explanations for the high prevalence of these mental health concerns is the finding of an association between time working with COVID-19 patients and

**Table 3. Results of multivariable regression model examining association between sociodemographic, work-related, psychosocial characteristics and the presence of depression according to CESD-10 cut-off score.**

| | % Yes | % No | Unadjusted OR (95% CI) | Adjusted OR (95%CI) |
|---|---|---|---|---|
| **Gender** | | | | |
| Male | 24 (15.1%) | 59 (24.0%) | 1.00 | 1.00 |
| Female | 135 (84.9%) | 187 (76.0%) | 1.78 (1.05–3.00) | 2.26 (1.00–5.10) * |
| **Age** | | | | |
| 20–35 | 75 (46.9%) | 100 (41.0%) | 1.00 | 1.00 |
| 36–50 | 55 (34.4%) | 94 (38.5%) | 0.78 (0.50–1.22) | 0.83 (0.40–1.75) |
| 51+ | 30 (18.8%) | 50 (20.5%) | 0.80 (0.47–1.38) | 0.98 (0.36–2.710) |
| **Relationship status** | | | | |
| Single | 66 (41.3%) | 100 (40.8%) | 1.00 | |
| In a relationship | 94 (58.8%) | 145 (59.2%) | 1.02 (0.68–1.53) | |
| **Level of education** | | | | |
| Undergraduate | 46 (29.3%) | 76 (31.9%) | 1.00 | |
| Bachelor's degree | 97 (61.8%) | 123 (51.7%) | 1.30 (0.83–2.05) | |
| Postgraduate Degree | 14 (8.9%) | 39 (16.4%) | 0.59 (0.29–1.21) | |
| **Place of work** | | | | |
| Primary healthcare clinics | 63 (39.9%) | 94 (39.5%) | 1.00 | |
| District hospitals | 16 (10.1%) | 22 (9.2%) | 1.09 (0.53–2.23) | |
| Tertiary hospitals | 79 (50.0%) | 122 (51.3%) | 0.97 (0.63–1.48) | |
| **Role at work** | | | | |
| Clinical | 114 (71.3%) | 195 (79.6%) | 1.00 | 1.00 |
| Managerial | 9 (5.6%) | 12 (4.9%) | 1.28 (0.52–3.14) | 3.28 (0.87–12.38) |
| Both | 37 (23.1%) | 38 (15.5%) | 1.67 (1.00–2.77) * | 2.17 (0.95–4.94) |
| **Fear of getting COVID-19** | | | | |
| No | 66 (41.8%) | 108 (46.4%) | 1.00 | |
| Yes | 92 (58.2%) | 125 (53.6%) | 1.20 (0.80–1.81) | |
| **Fear of spreading COVID-19** | | | | |
| No | 51 (32.3%) | 69 (29.6%) | 1.00 | |
| Yes | 107 (67.7%) | 164 (70.4%) | 0.88 (0.57–1.37) | |
| **Work time spent working with COVID-19 patients (med, range)** | 5.00 (10.00) | 3.00 (10.00) | 1.13 (1.05–1.22) * | 1.11 (0.99–1.24) |
| **COPE-R Self-distraction (med, range)** | 5.00 (6.00) | 5.00 (6.00) | 1.21 (1.07–1.38) * | 1.14 (0.91–1.43) |
| **COPE-R Active coping (med, range)** | 5.00 (6.00) | 5.00 (6.00) | 0.99 (0.88–1.11) | |
| **COPE-R Denial (med, range)** | 3.00 (6.00) | 2.00 (6.00) | 1.37 (1.17–1.60) * | 0.99 (0.75–1.31) |
| **COPE-R Substance use (med, range)** | 2.00 (6.00) | 2.00 (6.00) | 1.34 (1.12–1.59) * | 1.23 (0.94–1.61) |
| **COPE-R Emotional support (med, range)** | 5.00 (6.00) | 5.00 (6.00) | 1.02 (0.90–1.14) | |
| **COPE-R Instrumental support (med, range)** | 5.00 (6.00) | 4.00 (6.00) | 1.14 (1.01–1.28) * | 0.97 (0.78–1.21) |
| **COPE-R Behavioural disengagement (med, range)** | 3.00 (6.00) | 2.00 (5.00) | 1.81 (1.52–2.15) * | 1.50 (1.14–1.97) * |
| **COPE-R Venting (med, range)** | 5.00 (6.00) | 4.00 (6.00) | 1.55 (1.33–1.80) * | 1.06 (0.08–1.41) |
| **COPE-R Positive reframing (med, range)** | 5.00 (6.00) | 5.00 (6.00) | 1.03 (0.92–1.15) | |
| **COPE-R Planning (med, range)** | 5.00 (6.00) | 5.00 (6.00) | 1.18 (1.05–1.33) * | 1.13 (0.90–1.43) |
| **COPE-R Humour (med, range)** | 4.00 (6.00) | 3.00 (6.00) | 1.18 (1.05–1.33) * | 1.03 (0.84–1.26) |
| **COPE-R Acceptance (med, range)** | 6.00 (6.00) | 6.00 (6.00) | 1.09 (0.97–1.22) | |
| **COPE-R Religion (med, range)** | 6.00 (6.00) | 6.00 (6.00) | 1.05 (0.95–1.15) | |
| **COPE-R Self-blame (med, range)** | 3.00 (6.00) | 2.00 (6.00) | 1.80 (1.50–2.17) * | 1.29 (0.96–1.73) |
| **GHQ Intimate partner support** | | | | |
| No | 45 (29.6%) | 50 (21.7) | 1.00 | |
| Yes | 107 (70.4%) | 180 (78.3%) | 0.66 (0.41–1.06) | |

*(Continued)*

**Table 3.** (Continued)

| | % Yes | % No | Unadjusted OR (95% CI) | Adjusted OR (95%CI) |
|---|---|---|---|---|
| **GHQ Friend support** | | | | |
| No | 45 (28.8%) | 73 (31.5%) | 1.00 | |
| Yes | 111 (71.2%) | 159 (68.5%) | 1.13 (0.73–1.77) | |
| **GHQ Family support** | | | | |
| No | 41 (27.2%) | 52 (22.9%) | 1.00 | |
| Yes | 110 (72.8%) | 175 (77.1%) | 0.80 (0.50–1.28) | |
| **GHQ Mental health prof support** | | | | |
| No | 96 (61.5%) | 144 (62.6%) | 1.00 | |
| Yes | 60 (38.5%) | 86 (37.4%) | 1.05 (0.69–1.59) | |
| **GHQ Phoneline/self-help support** | | | | |
| No | 120 (79.9%) | 177 (76.3%) | 1.00 | |
| Yes | 36 (23.1%) | 55 (23.7%) | 0.97 (0.60–1.56) | |
| **GHQ GP/doc support** | | | | |
| No | 86 (55.1%) | 131 (57.0%) | 1.00 | |
| Yes | 70 (44.9%) | 99 (43.0%) | 1.08 (0.72–1.62) | |
| **GHQ Religious leader support** | | | | |
| No | 88 (57.1%) | 135 (58.2%) | 1.00 | |
| Yes | 66 (42.9%) | 97 (41.8%) | 1.04 (0.69–1.58) | |
| **Compassion satisfaction (ProQOL)** | | | | |
| Low/moderate | 110 (45.5%) | 132 (54.5%) | 1.00 | 1.00 |
| High | 45 (28.5%) | 113 (71.5%) | 0.48 (0.31–0.73) * | 0.57 (0.27–1.22) |
| **Secondary traumatic stress (ProQOL)** | | | | |
| Low | 61 (25.2%) | 181 (74.8%) | 1.00 | 1.00 |
| Moderate/high | 128 (80.5%) | 31 (19.5%) | 12.25 (7.52–19.96) * | 6.57 (3.30–13.04) * |
| **Burnout (ProQOL)** | | | | |
| Low | 86 (35.7%) | 155 (64.3%) | 1.00 | 1.00 |
| Moderate/high | 131 (82.9%) | 27 (17.1%) | 8.75 (5.35–14.29) * | 4.09 (1.91–8.75) * |

anxiety. Several studies that occurred during the COVID-19 period reported an increased risk of anxiety, the more time healthcare workers spent working with COVID-19 positive patients. This is not surprising, as spending more time working with COVID-19-positive patients increases healthcare workers' exposure to the stressors of the pandemic, such as the risk of infection, patient deaths, and inadequate personal protective equipment [52, 53]. Although rotation of staff may be used as a strategy to reduce this anxiety, staff shortages in such pandemic circumstances may make such an approach impossible to implement.

Third, several avoidant or maladaptive coping strategies were associated with experiencing symptoms of depression or anxiety. Only behavioural disengagement as a form of avoidant coping was associated with greater odds of depression among this sample of healthcare providers. On the other hand, substance use, venting and self-blame coping strategies were associated with greater odds of anxiety. The association between substance use coping and anxiety is not surprising given earlier local studies that have highlighted substance use as a way to numb or forget life stressors and manage traumatic stress [54, 55]. It appears that participants in our study interpreted the venting coping strategy as a form of complaining rather than seeking meaningful support. Such behaviour may be linked to rumination and could be the focus of further research. Additionally, self-blame was also found to be associated with anxiety; this is not surprising since there is a known association between self-blame and anxiety symptoms

**Table 4. Results of multivariable regression model examining association between sociodemographic, work-related, psychosocial characteristics and the presence of anxiety according to the GAD-7 cut-off scores.**

| | % Yes | % No | Unadjusted OR (95% CI) | Adjusted OR (95%CI) |
|---|---|---|---|---|
| **Gender** | | | | |
| Male | 34 (18.5%) | 47 (21.5%) | 1.00 | 1.00 |
| Female | 150 (81.5%) | 172 (78.5%) | 1.21 (0.74–1.97) | 0.88 (0.44–1.78) |
| **Age** | | | | |
| 20–35 | 87 (47.0%) | 87 (40.1%) | 1.00 | 1.00 |
| 36–50 | 63 (34.1%) | 88 (40.6%) | 0.72 (0.46–1.11) | 0.82 (0.43–1.58) |
| 51+ | 35 (18.9%) | 42 (19.4%) | 0.83 (0.49–1.43) | 1.23 (0.51–2.96) |
| **Relationship status** | | | | |
| Single | 79 (42.7%) | 85 (39.0%) | 1.00 | |
| In a relationship | 106 (57.3%) | 133 (61.0%) | 1.17 (0.78–1.74) | |
| **Level of education** | | | | |
| Undergraduate | 45 (24.7%) | 72 (34.1%) | 1.00 | |
| Bachelor's degree | 116 (63.7%) | 107 (50.7%) | 1.74 (1.10–2.74) | |
| Postgraduate degree | 21 (11.5%) | 32 (15.2%) | 1.05 (0.55–2.04) | |
| **Place of work** | | | | |
| Primary healthcare clinics | 67 (37.0%) | 91 (42.9%) | 1.00 | |
| District hospitals | 15 (8.3%) | 21 (9.9%) | 0.97 (0.47–2.02) | |
| Tertiary hospitals | 99 (54.7%) | 100 (47.2%) | 1.35 (0.88–2.05) | |
| **Role at Work** | | | | |
| Clinical | 136 (73.5%) | 171 (78.4%) | 1.00 | |
| Managerial | 11 (5.9%) | 11 (5.0%) | 1.26 (0.53–3.00) | |
| Both | 38 (20.5%) | 36 (16.5%) | 1.33 (0.80–2.21) | |
| **Fear of getting COVID-19** | | | | |
| No | 72 (39.6%) | 101 (49.0%) | 1.00 | |
| Yes | 110 (60.4%) | 105 (51.0%) | 1.47 (0.98–2.20) | |
| **Fear of spreading COVID-19** | | | | |
| No | 62 (30.1%) | 57 (31.3%) | 1.00 | |
| Yes | 144 (69.9%) | 125 (68.7%) | 0.94 (0.61–1.46) | |
| **Work time spent working with COVID-19 patients (med, range)** | 4.00 (10.00) | 3.00 (10.00) | 1.10 (1.03–1.18) * | 1.13 (1.02–1.25) * |
| **COPE-R Self Distraction (med, range)** | 5.00 (6.00) | 6.00 (6.00) | 1.17 (1.03–1.32) * | 1.11 (0.91–1.37) |
| **COPE-R Active coping (med, range)** | 5.00 (6.00) | 5.00 (6.00) | 0.98 (0.88–1.10) | |
| **COPE-R Denial (med, range)** | 2.00 (6.00) | 2.00 (6.00) | 1.31 (1.11–1.53) * | 0.98 (0.76–1.27) |
| **COPE-R Substance use (med, range)** | 2.00 (6.00) | 2.00 (6.00) | 1.50 (1.23–1.82) * | 1.39 (1.08–1.81) * |
| **COPE-R Emotional support (med, range)** | 5.00 (6.00) | 5.00 (6.00) | 1.00 (0.90–1.12) | |
| **COPE-R Instrumental support (med, range)** | 5.00 (6.00) | 4.00 (6.00) | 1.09 (0.97–1.22) | |
| **COPE-R Behavioural disengagement (med, range)** | 3.00 (6.00) | 2.00 (6.00) | 1.55 (1.31–1.84) * | 1.08 (0.84–1.39) |
| **COPE-R Venting (med, range)** | 5.00 (6.00) | 3.00 (6.00) | 1.67 (1.42–1.95) * | 1.31 (1.01–1.70) * |
| **COPE-R Positive reframing (med, range)** | 5.00 (6.00) | 5.00 (6.00) | 1.03 (0.93–1.15) | |
| **COPE-R Planning (med, range)** | 5.00 (6.00) | 5.00 (6.00) | 1.17 (1.04–1.31) * | 1.04 (0.85–1.27) |
| **COPE-R Humour (med, range)** | 4.00 (6.00) | 3.00 (6.00) | 1.14 (1.02–1.28) * | 0.92 (0.77–1.11) |
| **COPE-R Acceptance (med, range)** | 6.00 (6.00) | 6.00 (6.00) | 1.09 (0.97–1.23) | |
| **COPE-R Religion (med, range)** | 6.00 (6.00) | 6.00 (6.00) | 0.98 (0.89–1.08) | |
| **COPE-R Self-blame (med, range)** | 3.00 (6.00) | 2.00 (4.00) | 1.92 (1.58–2.34) * | 1.42 (1.08–1.87) * |
| **GHQ Intimate partner support** | | | | |
| No | 49 (27.5%) | 43 (21.2%) | 1.00 | |
| Yes | 129 (72.5%) | 160 (78.8%) | 0.71 (0.44–1.13) | |

*(Continued)*

**Table 4.** (Continued)

|  | % Yes | % No | Unadjusted OR (95% CI) | Adjusted OR (95%CI) |
|---|---|---|---|---|
| **GHQ Friend support** |  |  |  |  |
| No | 55 (30.4%) | 60 (29.4%) | 1.00 |  |
| Yes | 126 (69.6%) | 144 (70.6) | 0.96 (0.62–1.48) |  |
| **GHQ Family support** |  |  |  |  |
| No | 50 (28.4%) | 43 (21.4%) | 1.00 |  |
| Yes | 126 (71.6%) | 158 (78.6%) | 0.69 (0.43–1.10) |  |
| **GHQ Mental Health prof support** |  |  |  |  |
| No | 109 (60.2%) | 127 (62.9%) | 1.00 |  |
| Yes | 72 (39.8%) | 75 (37.1%) | 1.12 (0.74–1.69) |  |
| **GHQ Phoneline/self-help support** |  |  |  |  |
| No | 134 (74.0%) | 159 (77.9%) | 1.00 |  |
| Yes | 47 (26.0%) | 45 (22.1%) | 1.24 (0.78–1.98) |  |
| **GHQ GP/doc support** |  |  |  |  |
| No | 98 (54.4%) | 116 (57.1%) | 1.00 |  |
| Yes | 82 (45.6%) | 87 (42.9%) | 1.12 (0.75–1.67) |  |
| **GHQ Religious leader support** |  |  |  |  |
| No | 102 (57.3%) | 117 (57.1%) | 1.00 |  |
| Yes | 76 (42.7%) | 88 (42.9%) | 0.99 (0.66–1.49) |  |
| **Compassion satisfaction (ProQOL)** |  |  |  |  |
| Low/moderate | 124 (68.5%) | 117 (54.7%) | 1.00 | 1.00 |
| High | 57 (31.5%) | 97 (45.3%) | 0.55 (0.37–0.84) * | 0.84 (0.43–1.65) |
| **Secondary traumatic stress (ProQOL)** |  |  |  |  |
| Low | 47 (26.0%) | 163 (75.8%) | 1.00 | 1.00 |
| Moderate/high | 134 (74.0%) | 52 (24.2%) | 8.94 (5.67–14.10) * | 4.17 (2.25–7.71) * |
| **Burnout (ProQOL)** |  |  |  |  |
| Low | 44 (24.4%) | 135 (63.7%) | 1.00 | 1.00 |
| Moderate/high | 136 (75.6%) | 77 (36.3%) | 5.42 (3.49–8.42) * | 3.54 (1.82–6.99) * |

[56]. This phenomenon may occur when healthcare workers have witnessed multiple losses of patients, and possibly family members, during the pandemic. Further, as has been found in other studies [57, 58] they may have felt helpless in preventing these losses contributing to ongoing fear and anxiety about additional losses or infections in the hospital and at home.

Finally, high levels of secondary traumatic stress and burnout were found to increase the odds of both depression and anxiety symptoms. Similarly [59] found that secondary traumatic stress is associated with a range of negative outcomes such as, anxiety, depression, and decreased job satisfaction. Several studies conducted during the COVID-19 pandemic have confirmed this association [60, 61]. Interestingly, in this study, compassion satisfaction did not function as a protective factor. In contrast, studies have shown that compassion satisfaction can act as a protective factor for healthcare workers, helping them to cope with the stress and emotional toll of their work. When healthcare workers experience compassion satisfaction, it can help to mitigate the negative effects of burnout and compassion fatigue, both of which are common among healthcare workers [41].This discrepancy between the findings of this study and international literature, may be due to the abnormally high patient load and additional stressors associated with the pandemic, compassion satisfaction may not have the usual buffering effect against depressive symptoms.

The findings of this study need to be considered in the light of several limitations. First, given the cross-sectional nature of the study conducted in one province of South Africa, cause

and effect cannot be shown and the results may not be generalisable to the rest of the country or other LMIC settings However, data were collected from a wide range of healthcare facilities, and across different disciplines and two professions. Second, there are several other risk factors for depression and anxiety in this population that were not explored. Third, data were collected during different peaks of the COVID-19 pandemic which could have impacted on findings. South Africa enforced a rigorous "level 5" lockdown on March 27, 2020, disrupting research activities. Data collection was halted, and lockdown dates affected our ability to gather information. As lockdown levels eased, some research activities resumed with strict COVID-19 protocols. The prolonged lockdown finally ended on June 22, 2022. It's crucial to acknowledge that healthcare workers responding to surveys during different pandemic peaks may have experienced varying levels of distress due to different pandemic related factors like mortality rates. This timeframe highlights the need for careful interpretation of research findings during this unique period. Despite the limitations, this study provides valuable insight into the prevalence and risk factors associated with depression and anxiety highlighting a high prevalence of mental health issues among South African healthcare workers, and a clear unmet need for mental health care among these frontline workers. As such, there is a critical need to provide workplace mental health interventions to support healthcare workers. The WHO workplace guidelines for improving mental health and wellbeing in the workplace, highlights interventions which can be used at the individual, organizational, and societal levels [62]. Specifically, it may be helpful for the South African public health system to provide training and resources to help healthcare worker develop more effective coping strategies to support their wellbeing and to manage trauma, and burnout. Further, the presence of specific maladaptive coping strategies such as behavioural disengagement, venting and self-blame could be considered as markers of potential distress and also targeted in workplace interventions. Exploring the acceptability, feasibility, and effectiveness of implementing these interventions in the South African context could be a useful first step in addressing the mental health needs of healthcare workers.

## Acknowledgments

We are grateful to all the healthcare workers, working in healthcare facilities across the Western Cape, South Africa who voluntarily participated in the survey.

## Author Contributions

**Formal analysis:** Megan Pool, Katherine Sorsdahl, Claire van der Westhuizen.

**Supervision:** Claire van der Westhuizen.

**Writing – original draft:** Megan Pool.

**Writing – review & editing:** Katherine Sorsdahl, Bronwyn Myers, Claire van der Westhuizen.

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
