## [Decision Letter · Decision Letter 0]

17 Oct 2023

PONE-D-23-18610The prevalence of and factors associated with depressive and anxiety symptoms during the COVID-19 pandemic among healthcare workers in South Africa.PLOS ONE

Dear Dr. Pool,

Thank you for submitting your manuscript to PLOS ONE. After careful consideration, we feel that it has merit but does not fully meet PLOS ONE’s publication criteria as it currently stands. Therefore, we invite you to submit a revised version of the manuscript that addresses the points raised during the review process.

One of our associate editors and two reviewers carefully read the manuscript. Based on their evaluations the manuscript is major revision. The associate editor provided the following reasons:

The manuscript needs to be rewritten, taking into account the following main comments made by the reviewers

Introduction: restructuring of the text to provide more coherent and connected ideas and sections, including relevant references for this topic.

Methods: more details are needed.

Results: synthesize findings and present them in a systematic way.

Discussion:discuss main factors, secondary factors, generalizability, recommendations and implications

We look forward to receiving your revised manuscript.

Kind regards,

Juan Jesús García-Iglesias, Ph.D.

Academic Editor

PLOS ONE

Additional Editor Comments:

One of our associate editors and two reviewers carefully read the manuscript. Based on their evaluations the manuscript is major revision. The associate editor provided the following reasons:

The manuscript needs to be rewritten, taking into account the following main comments made by the reviewers

Introduction: restructuring of the text to provide more coherent and connected ideas and sections, including relevant references for this topic.

Methods: more details are needed.

Results: synthesize findings and present them in a systematic way.

Discussion:discuss main factors, secondary factors, generalizability, recommendations and implications

Reviewers' comments:

Reviewer's Responses to Questions

**Comments to the Author**

1. Is the manuscript technically sound, and do the data support the conclusions?

Reviewer #1: Yes

Reviewer #2: Yes

2. Has the statistical analysis been performed appropriately and rigorously? 

Reviewer #1: Yes

Reviewer #2: No

3. Have the authors made all data underlying the findings in their manuscript fully available?

Reviewer #1: Yes

Reviewer #2: Yes

4. Is the manuscript presented in an intelligible fashion and written in standard English?

Reviewer #1: Yes

Reviewer #2: Yes

5. Review Comments to the Author

Reviewer #1: Dear authors

I would like to thank you for giving me the opportunity to review the manuscript entitled “The prevalence of and factors associated with depressive and anxiety symptoms during the COVID-19 pandemic among healthcare workers in South Africa”. This study aimed to explore the prevalence and correlates of depression, anxiety, role-related stressors and coping strategies amongst healthcare workers during the COVID-19 pandemic in the Western Cape, South Africa. This study is well written and provide valuable information about mental health status of healthcare providers. I have some comments as follows:

- More literature review is required. The following article can be used for literature review and discussion of the findings:

1. https://www.cell.com/heliyon/pdf/S2405-8440(21)02673-6.pdf

2. https://www.frontiersin.org/articles/10.3389/fpubh.2022.1034624/full

- I suggest the “study sites” change to “study settings”

- How was the required sample size determined?

- More information is needed on the validity and reliability details of the instruments used.

- What ethical considerations were taken into account in data collection?

Reviewer #2: The paper reports about the prevalence and correlates of depression and anxiety among healthcare workers during the COVID-19 pandemic in the Western Cape, South Africa, exploring the role-related stressors and coping strategies.

Here are attached my suggestions to improve the paper

Section study sites

“fifteen sites took part in the study… Three tertiary hospitals were included, six district hospitals, three regional hospitals and eleven primary healthcare facilities”. It appears that 23 sites were included. Please clarify

Section participants

“…registered with either the Health Professions Council of South Africa (HPCSA) or the South African Nursing Council (SANC), and worked in either a clinical role as a nurse or doctor and/or a managerial position”. Authors mean “worked in either a clinical role and/or a managerial position as a nurse or a doctor?”. If this is correct, please reword.

Table 3

I suggest to exclude the column “no”, unless it conveys some important information that authors may want to explain. Also, I do not find it correct to adjust ORs including significant variables already included in the previous model in a second model that does not add any other variable. Third, the caption is misleading. I would suggest “Results of multivariable regression model examining association between sociodemographic, work-related, psychosocial characteristics and the presence of depression according to CESD-10 cut-off score”

The same goes for table 4

Final remarks

In my opinion, the main methodological issue of the study is the broad (18 months) enrolling period. Whilst I do not know the timing of COVID-19 waves in South Africa, I assume that health workers answering the survey on february 2020 were facing a very different level of distress of those filling the questionnaires in 2022, for both the acute vs prolonged stress, different preparation, different mortality rates, different severity of the pandemic and so on. Authors must discuss this limitation in light of the south African context during COVID-19.

6. PLOS authors have the option to publish the peer review history of their article (what does this mean?). If published, this will include your full peer review and any attached files.

Reviewer #1: No

Reviewer #2: **Yes: **Camilla Gesi

---

## [Author Response · Author response to Decision Letter 0]

19 Jan 2024

Thank you very much for taking the time to review our manuscript. We appreciate all the comments, suggestions and feedback. Please see below a response to each of the comments received. 

Editor: 

1. Introduction: restructuring of the text to provide more coherent and connected ideas and sections, including relevant references for this topic 

1. Thank you for the comment. We have done some re-structing and added link in sentences as well as references. 

2. Methods: more details are needed. 

2. Thank you for the comment. We have added details particularly ethical approval, study procedures and measures. 

3. Results: synthesize findings and present them in a systematic way. 

3. Thank you for the suggested comment. We have reviewed the results and added subheadings for easy navigation. Please advise if we need to consider shortening the tables and supplying them as supplementary files. 

4. Discussion: discuss main factors, secondary factors, generalizability, recommendations and implications Thank you for the comment. These key features of a discussion are included. Please see pages 17-19. 

5. Thank you for the comment. We have gone through to match the paper with the style requirements. 

6. We note that the grant information you provided in the ‘Funding Information’ and ‘Financial Disclosure’ sections do not match. When you resubmit, please ensure that you provide the correct grant numbers for the awards you received for your study in the ‘Funding Information’ section. 

6. Thank you for the comment. We made sure they match and included the declaration as per funders requirement. This is the funder users name and the abbreviation to identify the award. (BM-NHSP)

7. In your Data Availability statement, you have not specified where the minimal data set underlying the results described in your manuscript can be found. PLOS defines a study's minimal data set as the underlying data used to reach the conclusions drawn in the manuscript and any additional data required to replicate the reported study findings in their entirety. All PLOS journals require that the minimal data set be made fully available. For more information about our data policy, please see http://journals.plos.org/plosone/s/data-availability. "Upon re-submitting your revised manuscript, please upload your study’s minimal underlying data set as either Supporting Information files or to a stable, public repository and include the relevant URLs, DOIs, or accession numbers within your revised cover letter. For a list of acceptable repositories, please see http://journals.plos.org/plosone/s/data-availability#loc-recommended-repositories. Any potentially identifying patient information must be fully anonymized. Important: If there are ethical or legal restrictions to sharing your data publicly, please explain these restrictions in detail. Please see our guidelines for more information on what we consider unacceptable restrictions to publicly sharing data: http://journals.plos.org/plosone/s/data-availability#loc-unacceptable-data-access-restrictions. Note that it is not acceptable for the authors to be the sole named individuals responsible for ensuring data access. We will update your Data Availability statement to reflect the information you provide in your cover letter.

7. The data that support these findings are owned by the Western Cape Department of Health and applicants may apply online to the National Health Research Database (https://nhrd.hst.org.za/).

8. Please include your full ethics statement in the ‘Methods’ section of your manuscript file. In your statement, please include the full name of the IRB or ethics committee who approved or waived your study, as well as whether or not you obtained informed written or verbal consent. If consent was waived for your study, please include this information in your statement as well. 

8. Ethical approval was obtained from the University of Cape Town Human Research Ethics Committee. In addition, ethical approval was obtained from the Western Cape Department of Health as well as the respective operational managers, and heads of departments for each hospital or clinic. Written informed consent was obtained for each participant. Please see page 5 of the manuscript. 

Reviewer 1: 

1. More literature review is required. The following article can be used for literature review and discussion of the findings:

a. https://www.cell.com/heliyon/pdf/S2405-8440(21)02673-6.pdf

b.https://www.frontiersin.org/articles/10.3389/fpubh.2022.1034624/full

1. Thank you for the comment. We have reviewed the introduction and discussion carefully; these papers are really interesting but since we focused on quantitative findings specifically systematic reviews and meta-analysis it did not seem appropriate to cite these papers here. However, we have qualitative findings from the broader study and these papers will be very helpful in the discussion of the data. 

2. I suggest the “study sites” change to “study settings”

2. Thank you for the suggestion, we have made this change. Please see page 5 of the manuscript. 

3. How was the required sample size determined? 

3. Thank you for the comment. We based our sample size calculation on the primary hypothesis those people experiencing burnout or secondary traumatic stress will be more likely to have depression. We set the power at 80% and the significance level at 0.05. Based on the literature we used a conservative estimate that 30% of those experiencing secondary traumatic stress or burnout would be experiencing high levels of depressive symptoms. We used open Epi to calculate the sample size for this cross-sectional study. For adequate power, it was determined that a total of 291 healthcare workers would be sufficient for this cross-sectional study. Please see page 8 & 9 of the manuscript. 

4. More information is needed on the validity and reliability details of the instruments used.

4. Thank you for this comment we had added details to the measures section. Please see page 7 & 8 of the manuscript. 

5. What ethical considerations were taken into account in data collection? 

5. Thank you for the comment. The following ethical considerations were taken into account during data collection. Participants were afforded privacy as the questionnaire was self-administered; participants could complete it in private at their own time. We protected confidentiality- collected completed forms in bulk and did not ask for names or contact details unless the participants wanted to take part in qualitative interview as part of the second phase of the study. We did not report any findings by clinic thus maintaining confidentiality of the participants. Participation in the study was voluntary and this was stressed in consent forms and with clinic management. Please see page 19 of the manuscript. 

Reviewer 2: 

1. Section study sites

“fifteen sites took part in the study… Three tertiary hospitals were included, six district hospitals, three regional hospitals and eleven primary healthcare facilities”. It appears that 23 sites were included. Please clarify

1. Thank you for this comment, we have rectified the manuscript to reflect that 23 sites took part in the study, please see page 5 of the manuscript. 

2. Section participants

“…registered with either the Health Professions Council of South Africa (HPCSA) or the South African Nursing Council (SANC) and worked in either a clinical role as a nurse or doctor and/or a managerial position”. Authors mean “worked in either a clinical role and/or a managerial position as a nurse or a doctor?”. If this is correct, please reword. 

2. Thank you for the suggested comment, we have made the relevant changes. Please see page 5 of the manuscript. 

3. Table 3

I suggest to exclude the column “no”, unless it conveys some important information that authors may want to explain. Also, I do not find it correct to adjust ORs including significant variables already included in the previous model in a second model that does not add any other variable. Third, the caption is misleading. I would suggest “Results of multivariable regression model examining association between sociodemographic, work-related, psychosocial characteristics and the presence of depression according to CESD-10 cut-off score”

3. Thank you for the suggested comments. First, we have changed the caption of the table to the suggested heading. Please see page 11 of the manuscript. Second, although we can see the rationale for excluding the “no” column, in our experience this column is helpful for our readers. Particularly since we used column percent for the two columns, and it may be helpful to see the distribution of variable in those who are not experiencing high levels of depressive symptoms. The use of statistically significant variable from the unadjusted regression models is a convention that is used in some disciplines, and we applied this as a method to reduce the number of variables in the adjusted model and avoid the pitfalls associated with stepwise regression. Additionally, the variable identified using this method were represented plausible associations identified in previous literature, Field, A. (2009) Discovering Statistics Using SPSS. 3rd Edition, Sage Publications Ltd., London pg. 213-214. 

4. The same goes for table 4

4. Thank you for the comment, please see response above. We have changed the title of the table please see page 15 of the manuscript. 

5. In my opinion, the main methodological issue of the study is the broad (18 months) enrolling period. Whilst I do not know the timing of COVID-19 waves in South Africa, I assume that health workers answering the survey on february 2020 were facing a very different level of distress of those filling the questionnaires in 2022, for both the acute vs prolonged stress, different preparation, different mortality rates, different severity of the pandemic and so on. Authors must discuss this limitation in light of the south African context during COVID-19 

5. Thank you for the comment. We agree that this is a limitation- give timing of the lockdown periods. We have added this as limitation as per your suggestions. Please see page 19 of the manuscript.

---

## [Decision Letter · Decision Letter 1]

13 Feb 2024

The prevalence of and factors associated with depressive and anxiety symptoms during the COVID-19 pandemic among healthcare workers in South Africa.

PONE-D-23-18610R1

Dear Dr. Pool,

We’re pleased to inform you that your manuscript has been judged scientifically suitable for publication and will be formally accepted for publication once it meets all outstanding technical requirements.

Kind regards,

Juan Jesús García-Iglesias, Ph.D.

Academic Editor

PLOS ONE

Additional Editor Comments (optional):

I have reviewed the changes made and I am pleased to see that you have addressed the comments and suggestions in a very satisfactory manner. The manuscript has significantly improved.

Reviewers' comments:

Reviewer's Responses to Questions

**Comments to the Author**

1. If the authors have adequately addressed your comments raised in a previous round of review and you feel that this manuscript is now acceptable for publication, you may indicate that here to bypass the “Comments to the Author” section, enter your conflict of interest statement in the “Confidential to Editor” section, and submit your "Accept" recommendation.

Reviewer #2: All comments have been addressed

2. Is the manuscript technically sound, and do the data support the conclusions?

Reviewer #2: Yes

3. Has the statistical analysis been performed appropriately and rigorously? 

Reviewer #2: Yes

4. Have the authors made all data underlying the findings in their manuscript fully available?

Reviewer #2: Yes

5. Is the manuscript presented in an intelligible fashion and written in standard English?

Reviewer #2: Yes

6. Review Comments to the Author

Reviewer #2: (No Response)

7. PLOS authors have the option to publish the peer review history of their article (what does this mean?). If published, this will include your full peer review and any attached files.

Reviewer #2: **Yes: **Camilla Gesi

---

## [Editor Report · Acceptance letter]

26 Feb 2024

PONE-D-23-18610R1 

PLOS ONE

Dear Dr. Pool, 

I'm pleased to inform you that your manuscript has been deemed suitable for publication in PLOS ONE. Congratulations! Your manuscript is now being handed over to our production team.

Kind regards, 

on behalf of

Dr. Juan Jesús García-Iglesias 

Academic Editor

PLOS ONE